# From Pathways to Practice: Impact of Implementing Mobilization Recommendations in Head and Neck Cancer Surgery with Free Flap Reconstruction

**DOI:** 10.3390/cancers13122890

**Published:** 2021-06-09

**Authors:** Rosie Twomey, T. Wayne Matthews, Steven C. Nakoneshny, Christiaan Schrag, Shamir P. Chandarana, Jennifer Matthews, David McKenzie, Robert D. Hart, Na Li, Joseph C. Dort, Khara M. Sauro

**Affiliations:** 1Ohlson Research Initiative, Arnie Charbonneau Research Institute, University of Calgary Cumming School of Medicine, 3280 Hospital Dr NW Calgary, Calgary, AB T2N 4Z6, Canada; rosemary.twomey@ucalgary.ca (R.T.); wmatthew@ucalgary.ca (T.W.M.); scnakone@ucalgary.ca (S.C.N.); cschrag@ucalgary.ca (C.S.); shamir.chandarana@ucalgary.ca (S.P.C.); jennifer.matthews2@ucalgary.ca (J.M.); charles.mckenzie@albertahealthservices.ca (D.M.); robert.hart@albertahealthservices.ca (R.D.H.); jdort@ucalgary.ca (J.C.D.); 2O’Brien Institute of Public Health, Cumming School of Medicine, University of Calgary, 3280 Hospital Dr NW Calgary, Calgary, AB T2N 4Z6, Canada; 3Section of Otolaryngology Head & Neck Surgery, Department of Surgery, University of Calgary Cumming School of Medicine, 3280 Hospital Drive NW, Calgary, AB T2N 4Z6, Canada; 4Foothills Medical Centre, Alberta Health Services, 1403 29 St NW, Calgary, AB T2N 2T9, Canada; 5Section of Plastic and Reconstructive Surgery, Department of Surgery, University of Calgary Cumming School of Medicine, 3330 Hospital Dr NW, Calgary, AB T2N 4N1, Canada; 6Department of Community Health Sciences, Surgery & Oncology, University of Calgary Cumming School of Medicine, 3D10, 3280 Hospital Drive NW Calgary, Calgary, AB T2N 4Z6, Canada; Na.Li@ucalgary.ca

**Keywords:** evidence-based medicine, head and neck surgery, clinical practice guidelines, care pathways, clinical pathways, enhanced recovery, early mobilization, clinical outcomes improvement, quality improvement, implementation science, registry, administrative data

## Abstract

**Simple Summary:**

Treatment for head and neck cancer (HNC) often involves complex surgery to remove the tumour followed by a reconstructive procedure to restore function and appearance. Getting out of bed and moving after surgery (early mobilization) is key to a good recovery. Clinical guidelines (called Enhanced Recovery after Surgery or ERAS guidelines) recommend getting out of bed and moving in the first 24 h after HNC surgery. This study looks at compliance to mobilization recommendations in 445 patients within an ERAS care pathway for HNC surgery. Implementing a new mobilization recommendation resulted in a 10% increase in recommendation compliance, despite a more aggressive target for (from 48 to 24 h). Patients who had surgery after the new guideline were more likely to leave the hospital on time (within ten days after surgery). Engaging the healthcare team and changing the care instructions improved mobilization and adherence to guideline-recommended care after HNC surgery with free flap reconstruction.

**Abstract:**

One of the foundational elements of enhanced recovery after surgery (ERAS) guidelines is early postoperative mobilization. For patients undergoing head and neck cancer (HNC) surgery with free flap reconstruction, the ERAS guideline recommends patients be mobilized within 24 h postoperatively. The objective of this study was to evaluate compliance with the ERAS recommendation for early postoperative mobilization in 445 consecutive patients who underwent HNC surgery in the Calgary Head and Neck Enhanced Recovery Program. This retrospective analysis found that recommendation compliance increased by 10% despite a more aggressive target for mobilization (from 48 to 24 h). This resulted in a decrease in postoperative mobilization time and a stark increase in the proportion of patients mobilized within 24 h (from 10% to 64%). There was a significant relationship between compliance with recommended care and time to postoperative mobilization (Spearman’s *rho* = −0.80; *p* < 0.001). Hospital length of stay was reduced by a median of 2 days, from 12 (1QR = 9–16) to 10 (1QR = 8–14) days (*z* = 3.82; *p* < 0.001) in patients who received guideline-concordant care. Engaging the clinical team and changing the order set to support clinical decision-making resulted in increased adherence to guideline-recommended care for patients undergoing major HNC surgery with free flap reconstruction.

## 1. Background and Rationale

Clinical practice guidelines are systematically developed syntheses of the best available evidence that incorporate expert opinion through consensus and help guide clinical care [1]. The goal of clinical practice guidelines is to increase high-quality care and reduce inappropriate interventions [2]. When clinical practice guidelines are implemented effectively, they have been shown to improve clinical outcomes [3]. However, uptake and incorporating evidence-based practice guideline recommendations in clinical practice can be challenging [4,5]. Care that is not recommendation compliant can contribute to unwarranted clinical variation and can compromise the quality of care [6]. Implementation science, the application and integration of evidence into practice, has made great advances to understanding why guidelines are not readily adopted into clinical practice [7]. Guidelines that are developed by sources perceived as credible, that are applicable to the target population, aligned with current views of best practice, and are supported by organizational structures are more likely to be adopted into clinical practice [5,8,9].

The Enhanced Recovery After Surgery (ERAS) Society is an international society dedicated to developing clinical practice guidelines to reduce unwarranted variation and improve the quality of care for surgical patients [10,11]. The ERAS guidelines adopt a multidisciplinary and multimodal approach and address preoperative, intraoperative, and postoperative care elements to optimize outcomes after surgery. An ERAS guideline includes consensus recommendations based on a literature review and quality assessment [10]. ERAS guidelines are developed and tailored for specific types of surgery, including major head and neck cancer (HNC) surgery with microvascular free flap reconstruction [12].

One of the foundations of ERAS is early postoperative mobilization [11]. Early mobilization prevents or reduces the detrimental impact of prolonged bed rest and immobility after surgery and is, therefore, a recommendation in all ERAS guidelines, including several specific to surgical oncology [12,13,14,15]. In HNC surgery with free flap reconstruction, ERAS strongly recommends that patients are mobilized within 24 h [12], based on data from one HNC-specific cohort study [16] and evidence from other types of surgeries [17]. In our companion paper, we found that mobilization delayed beyond 24 h was associated with more complications, providing novel HNC-specific evidence for the ERAS recommended timeframe [18]. However, compliance with early mobilization recommendations can be low, particularly in the early adoption of an enhanced recovery pathway [19].

In addition to guiding clinical care, an essential component of ERAS is audit and evaluation. Audit and evaluation of ERAS guidelines involve assessments of compliance of clinical practice to the recommended care; that is, how well care elements have been implemented into clinical practice. Some individual care elements may be more strongly associated with improved outcomes than others, but increased compliance with the entire ERAS protocol is also associated with further improvements [20]. Few studies have reported compliance with ERAS recommendations for early mobilization in major HNC surgery [19,21,22,23,24] or the impact of delivering recommendation compliant care on outcomes. The objective of this study was to evaluate compliance with the ERAS recommendation for early postoperative mobilization after a change in the recommendation and the impact of recommendation compliant care on postoperative complications and hospital length of stay (LOS) among patients undergoing major HNC surgery with free flap reconstruction within the context of a long-term quality management program [25]. 

## 2. Materials and Methods

The materials and methods have been previously described in detail in our companion paper [18]. 

### 2.1. Setting and Participants

Briefly, all surgeries were performed at Foothills Medical Centre, the tertiary referral centre for major HNC surgery with free flap reconstruction in Southern Alberta, Canada. Patients were part of the Calgary Head and Neck Enhanced Recovery Pathway, which includes an established measurement, audit, and feedback system [26]. The care pathway is fully integrated into the inpatient electronic medical record, and clinical data are prospectively collected by a trained research assistant who is embedded in the hospital inpatient unit [25]. Consecutive adult patients undergoing HNC resection with free flap reconstruction between 4 June 2012, and 31 September 2020, were included in this analysis. Resection with free flap reconstruction for head and neck malignancies, benign tumours, or complications of other treatments (e.g., osteoradionecrosis) were included. Patients were excluded if data for postoperative mobilization (pathway data) were missing and if they were aged <18 years old. 

### 2.2. Early Mobilization Recommendations

The Calgary program uses a formal care pathway, integrated into clinical care using a computerized order set, to deliver care to the target patient population. A formal measurement system, including audit and feedback, is part of the program, and formal quality reports are generated and reviewed by the unit quality council. This council meets regularly and includes nurses, allied health professionals, surgeons, and other clinicians responsible for care delivery. A review of the reports in 2015 revealed sub-optimal mobilization times, which was discussed at a quality council meeting. Potential reasons were explored, and the team concluded that three factors were of major importance: (1) The current pathway (at that time) only required mobilization within 48 h (postoperative day; POD 2). (2) The current pathway specified the foley catheter be removed on POD 3. (3) Unit staff were not sufficiently aware of the importance of early mobilization and, therefore, might be paying insufficient attention to it. 

As a result, the team developed a multifaceted approach to address the mobilization issue as follows:(1)The pathway and order set were modified to specify mobilization within 24 h instead of within 48 h (POD 2).(2)The pathway and order set were modified to specify foley catheter removal on POD 2 at the latest instead of POD 3.(3)Formal education sessions for unit nursing staff were developed and implemented. These sessions provided knowledge outlining the importance of early mobilization on postoperative recovery, and all unit staff attended the sessions.

These three interventions were integrated into the existing pathway in February of 2016. Therefore, there were two study periods: (1) 4 June 2012, to 3 February 2016, when the mobilization recommendation was within 48 h; and (2) 4 February 2016, to 30 September 2020, when the mobilization recommendation was within 24 h.

In December 2017, we also implemented two additional care elements to make our program consistent with the published ERAS guidelines. The additional care elements were an intraoperative fluid management protocol and a perioperative multimodal analgesia protocol. 

### 2.3. Variables

Early mobilization: The date of first meaningful mobilization was recorded as the date where there was evidence that the patient was mobilized out of bed, up in a chair, standing and/or walking. Time to mobilization was calculated as the number of calendar days from the date of surgery (POD 0) to the date of first meaningful mobilization (evidence that the patient was mobilized out of bed, up in a chair, standing, and/or walking). Postoperative early mobilization was also categorized as POD 0–1 (a surrogate for mobilization within 24 h postoperative, in line with current pathway recommendations [12]) vs. POD >1 (mobilization after 24 h postoperative). Postoperative mobilization was also categorized as POD 0–2 (a surrogate for within 48 h) and POD >2 (mobilization after 48 h postoperative).

Recommendation compliance: we evaluated compliance with the mobilization recommendation as a dichotomous variable (yes, no). Compliance with recommended care was defined as meaningful mobilization by POD 2 before the change in recommendation (before June 2016) and POD 1 after the change in recommendation (June 2016 onward). In other words, adherence is the proportion of patients who received guideline-concordant care. 

Postoperative complications: we defined postoperative complications in two ways: (1) occurrence of any complication (yes, no) was defined as any deviation from the normal postoperative course and was classified as grade 1-V using the Clavien–Dindo classification [27], and (2) major complications (yes, no) were defined as grade IIIb–V using the Clavien–Dindo classification, which includes complications requiring surgical, endoscopic or radiological intervention under general anesthesia, life-threatening complications, and death [27]. In separate analyses, we also used dichotomous variables (yes, no) to examine specific complication types, including pneumonia, pulmonary embolism, deep vein thrombosis, delirium tremens, myocardial infarction, bleed or hematoma, free flap compromise and failure. 

Length of stay: LOS was calculated as the time interval from the date of surgery (postoperative day zero) and the date of discharge. LOS was reported as a continuous variable (median and IQR) and was dichotomized as postoperative day 0–10 and after postoperative day 10 (the latter indicating a prolonged LOS).

### 2.4. Statistical Methods

All data were analyzed using Stata 16.1 (Stata Corp, College Station, TX, USA) [28] and alpha was set at 0.05 for all statistical tests. To describe patient demographic and clinical characteristics and the proportion of patients mobilized according to pathway recommended timeframes, data were summarized as frequencies (percentages) for categorical variables, median (interquartile range, IQR) or mean (standard deviation, SD), and range for continuous variables. To compare demographic and clinical characteristics between patients who underwent surgery before vs. after the pathway recommendation change, two-tailed Fisher’s exact tests for r × c contingency tables were used for categorical variables [29], an independent t-test was used for age on the day of surgery, and Mann–Whitney tests were used for time to mobilization and LOS (where reported as continuous outcomes). 

A Spearman rank correlation (time to postoperative mobilization) and logistic regression (major postoperative complications and LOS) were used to explore the association between outcomes and compliance with recommended care (exposure). We also explored whether recommendation compliance (yes/no) was a significant predictor of major postoperative complications and a prolonged LOS using multivariable models presented in our companion paper. Procedures for variable selection for our previous logistic regression models to identify predictors of postoperative complications and LOS are described in detail in our companion paper [18]. 

## 3. Results

Postoperative mobilization data were missing in 11% (*n* = 55) of cases. It is believed the missing data were missing at random either due to failure to flag a case for data collection or a failure to record mobilization milestones in the chart [18].

A total of 445 patients were included in this analysis. Patients were aged 61.2 ± 12.2 years (mean ± SD), and the majority of patients were men (68%) with at least one comorbidity (63%). Most patients were diagnosed with stage III–IV (65%) squamous cell carcinoma (80%), and the primary site was most commonly the oral cavity (68%). Fewer (*n* = 164, 37%) patients underwent surgery before the change in mobilization recommendation than after (*n* = 281, 63%) the change in mobilization recommendation. Characteristics of patients in the two study periods are presented and compared in Table 1. Patients who underwent surgery after the change in mobilization recommendation more frequently reported light–moderate drinking, and fewer patients reported heavy drinking behaviours in comparison to those who underwent surgery before the change in mobilization recommendation (Table 1).

### 3.1. Recommendation Compliance

Overall compliance with the recommended care was 60%. Compliance with recommended care increased from 54% before the change (mobilization within 48 h) to 64% after the change (mobilization within 24 h) (OR = 1.52; 95% CI = 1.00–2.28; *p* = 0.045). The change in the mobilization recommendation and increased compliance with guideline-concordant care resulted in a stark increase in the proportion of patients mobilized within 24 h (from 10% to 64%). Patients were 15.2 times more likely to be mobilized within 24 h when the recommendation was introduced (95% CI = 8.5–28.4; *p* < 0.001). The decrease in postoperative mobilization time is presented in Figure 1 and Appendix A.

### 3.2. The Association between Compliance with Recommended Care and Postoperative Complications and LOS

There was a significant relationship between compliance with recommended care and time to postoperative mobilization (Spearman’s *rho* = −0.80; *p* < 0.001). Controlling for alcohol status alone (Table 1), compliance with recommended care meant that patients were less likely to have pneumonia (OR = 0.35; 95% CI = 0.16–0.75; *p* = 0.007 model AIC = 210), less likely to have a major complication (OR = 0.44; 95% CI = 0.25–0.76; *p* = 0.004; model AIC = 338), and less likely to have a prolonged LOS (OR=0.58; 95% CI = 0.38–0.89; *p* = 0.013; model AIC = 529; Appendix A). More generally, LOS was reduced by a median of 2 days, from 12 (1QR = 9–16) to 10 (1QR = 8–14) days (*z* = 3.82; *p* < 0.001) when patients received guideline-concordant care. We also explored whether compliance with the recommendation (yes/no) was a significant predictor within our multivariable models of major postoperative complications and LOS (presented in our companion paper). Taking other important covariates into consideration (including alcohol status), we previously found that mobilization after 48 h was associated with having a major postoperative complication and a prolonged LOS^18^. Here, we found that compliance with the mobilization recommendation was not itself a predictor within these models (when substituting mobilization within 48 h for compliance, considering the overlap in these variables; Appendix A).

## 4. Discussion

This study showed that implementing the ERAS-concordant recommendations for early mobilization resulted in a 10% increase in compliance with recommended care despite a more aggressive target for mobilization (from 48 to 24 h). There was a corresponding stark increase in the proportion of patients mobilized early after HNC surgery with free flap reconstruction (from 10 to 64%). In patients who received guideline-concordant care, hospital LOS was reduced by two days, and patients were less likely to experience a prolonged LOS, pneumonia, or a major complication. Taken together, our findings suggest that it was not likely the moderate improvement in recommendation compliance but rather the substantial improvement in early mobilization that was responsible for the improvement in these outcomes.

In the present study, we found an improvement in early mobilization in response to the change in the guideline-recommended mobilization time. This increase in the proportion of patients mobilized within 24 h by >50% was unlikely due to differences in patient acuity between patients in the two study periods. Instead, factors related to changing and implementing the guideline were likely to have played a key role in mobilization time. Including stakeholder engagement in the process to identify and address barriers to mobilizing patients within the recommended 24 h, adding prompts and clinical decision-making tools to the pathway and education around early mobilization are likely some of the most important factors that impacted mobilization time. Indeed, these factors are known to be predictors of guideline adoption [5,8,9]. Taking a multidimensional approach to complex interventions, as done by our group, has also been found to be successful for providing care that is compliant with recommended care [30]. The audit and feedback system was also important to support efficient and effective changes to the guideline and facilitating the delivery of recommendation-compliant care. Audit and feedback can be a moderately effective intervention to change clinical practice [31]. Audit and feedback are especially effective when baseline performance is poor and when the feedback is provided by a respected colleague or superior [31] as was the case in the current study. It is likely that all of the factors discussed contributed to the successful implementation and compliance with the guideline; however, there was not a systematic implementation process and evaluation; therefore, we cannot be certain this is the case. We advocate for conducting formal implementation studies, including process evaluations, when implementing changes to clinical practice to identify factors that contributed to changes in clinical practice and outcomes. Such studies can inform and streamline the implementation of changes in clinical practice in the future.

Multicomponent enhanced recovery pathways are effective at reducing overall complications and LOS, but there is substantial heterogeneity, and the exact nature of the successful intervention can be difficult to establish [32]. One source of heterogeneity is fidelity to the entire pathway through the adoption of each element into clinical practice. Subsequent to the publication of the 2017 ERAS guideline [12], studies have reported compliance with early mobilization recommendations and clinical practice for major HNC, with mixed results [19,21,22,23,24]. Following the initial development and implementation of an ERAS, Coyle et al. reported that only 7% of patients were mobilized within 24 h [19]. In contrast, Low et al. (2020) recently reported 56.7% compliance to mobilization recommendations within 24 h (88.5% within 48 h) [21] after implementing an ERAS pathway with a default postoperative order set for mobilization on POD 1. Including physical and occupational consults on POD 1 as part of the order set, while limited by local resources, may also be an important strategy to increase adherence to mobilization recommendations [21,22]. The highest reported proportion of patients mobilized within 24 h after HNC surgery with free flap reconstruction is 86% (97% within 48 h) [22]. Part of the pathway described by Imai et al. (2020) can be attributed to the preoperative guidance on mobilization delivered by a physical therapist. Patients were encouraged to walk before surgery and be prepared for early mobilization after surgery, and a pedometer was incorporated from the beginning of hospitalization [22]. We have previously highlighted that providing patients with continuous monitoring of steps is an important avenue for future research on improving outcomes after HNC surgery [33].

Postoperative care of HNC patients takes place on a specialized ward—the hospital stay is resource-intensive and a significant economic burden [34,35,36]. The median decrease in LOS of two days when patients were mobilized according to guideline-concordant care was not only statistically significant but is also clinically meaningful. A two-day decrease in LOS results in a reduction of ~17% of the total LOS for these patients, which reduces overall care costs. Previous studies have shown that increases in care efficiency, such as reduced LOS, do not compromise care quality or patient satisfaction [35,36]. Alongside early mobilization, the ERAS guideline for HNC includes recommendations for perioperative nutritional care, pharmacologic thromboprophylaxis, perioperative antibiotics, postoperative flap monitoring, and tracheostomy care [12]. These care elements were already integrated into the Calgary Head and Neck Enhanced Recovery Pathway, and protocols were stable across the study period. However, two additional care elements were added in December 2017 when the ERAS guideline was published: perioperative multimodal analgesia protocol and an intraoperative fluid management protocol. The benefit of goal-directed fluid therapy may be marginal [37], but adequate pain control is expected to facilitate early mobilization. We have previously shown that the introduction of an opioid-sparing pain management protocol increased the proportion of patients with adequate pain control (pain rating below 4/10) by ~10% in the first 24 h after surgery [38]. Changes in pain management in our study could have contributed to compliance with the pathway recommendation, but it cannot account for the decrease in time to mobilization after the introduction of the new mobilization recommendation in February 2016 (Figure 1). However, we were unable to measure this in the present study, and we acknowledge this as a limitation.

Some additional limitations with this dataset have been highlighted previously, including a relative lack of data on the timing of all complications [18]. We also acknowledge that adherence to the ERAS recommendation only indicates the rapidity of mobilization and not the quantity (distance or step count) or total time spent mobilizing in the first 24 h after surgery. However, a limitation specific to the present analysis is that there were some unknowns in the dissemination of the evidence and implementation of the pathway recommendation. As more evidence on the value and safety of care is generated in this unique surgical discipline, new recommendations will need to be implemented (or previous recommendations de-implemented). Resources should be dedicated to high-quality implementation research to systematically study the process of translating evidence into clinical practice in HNC surgical care, as well as the impact of the new practice on relevant outcomes. Implementation science in perioperative care is a growing area of research [39] that has received limited attention in HNC surgery with free flap reconstruction. To narrow the evidence–practice gap, future additions or revisions of ERAS care elements should take an implementation science approach. This will help elucidate the factors influencing compliance with guideline-recommended care to improve patient outcomes.

## 5. Conclusions

In an enhanced recovery program for patients undergoing major HNC surgery, the proportion of patients receiving evidence-based, recommended care and early mobilization increased following the adoption of a new mobilization recommendation. Changing the order set to support clinical decision-making, engaging the clinical team, and creating a culture of mobilization resulted in increased compliance guideline-recommended care. Effectively implementing ERAS recommendations for early mobilization may contribute to a reduction in postoperative complications and reduced LOS. 

## Figures and Tables

**Figure 1 cancers-13-02890-f001:**
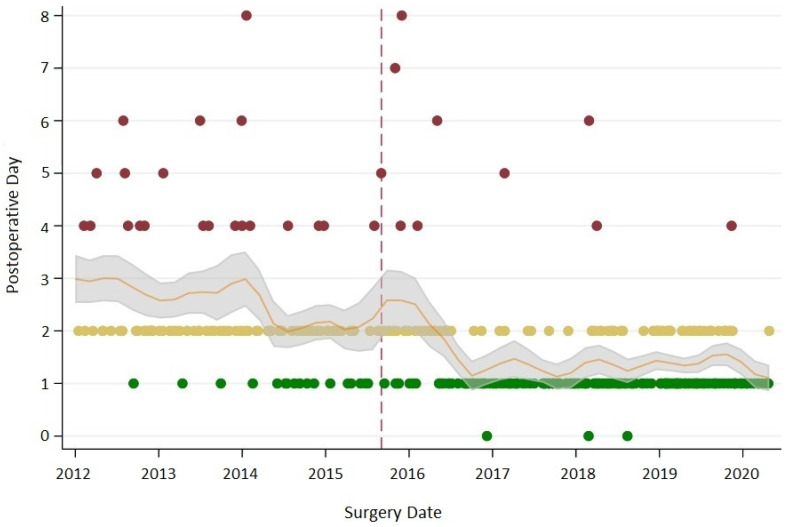
Mobilization time (Postoperative Day) across time. Yellow circles = mobilization on POD 2. Red circles = mobilization after POD 2. The dashed line indicates the change in the pathway recommendation, which is followed by a sharp decrease in the time to postoperative mobilization and an increase in the number of patients mobilized within 24 h (POD 0–1, green circles). The line of best fit (local polynomial smoother) with 95% CI is displayed for time to mobilization (continuous variable). For presentation purposes, data points (*n* = 9) for patients mobilized after POD 8 were removed.

**Table 1 cancers-13-02890-t001:** Patient characteristics before and after the mobilization recommendation across study periods.

Characteristic	All Cases *n* = 445	Before *n* = 164	After *n* = 281	*p*-Value
Sex				
Male	303 (68)	111 (68)	192 (68)	0.916
Female	142 (32)	53 (32)	89 (32)	
Age (years)				
Mean ± SD	61.2 ± 12.2	61.2 ± 11.6	61.2 ± 12.5	0.495 *
Range	21.2−89.0			
Alcohol status				0.027
Never	90 (20)	28 (17)	62 (22)	
Light/Moderate	162 (36)	47 (29)	115 (41)	
Heavy	93 (21)	44 (27)	49 (17)	
Former	48 (11)	17 (10)	31 (11)	
Not reported	52 (12)	28 (17)	24 (9)	
Smoking status				0.304
Never smoked	117 (26)	35 (21)	82 (29)	
Former smoker	151 (34)	58 (35)	93 (33)	
Current smoker	136 (31)	51 (31)	85 (30)	
Not reported	41 (9)	20 (12)	21 (8)	
Comorbidities				0.953
None	142 (32)	53 (32)	89 (32)	
One	136 (31)	51 (31)	85 (30)	
Two or more	167 (37)	60 (37)	107 (38)	
Specific Comorbidity(Present)				
Diabetes	54 (12)	20 (12)	34 (12)	1.000
COPD	50 (11)	23 (14)	27 (10)	0.164
Hypertension	181 (41)	66 (40)	115 (41)	0.921
Heart disease	59 (13)	24 (15)	35 (12)	0.563
Primary site				
Oral cavity	303 (68)	111 (68)	192 (68)	0.202
Pharynx and larynx	42 (8)	16 (10)	26 (9)	
Skin	39 (9)	9 (5)	18 (6)	
Paranasal/Nasal	27 (6)	10 (6)	29 (10)	
Other	34 (8)	18 (11)	16 (6)	
Histology				0.152
Squamous cell	356 (80)	128 (78)	228 (81)	
Other cancer	83 (18)	30 (18)	46 (16)	
Benign	6 (1)	0 (0)	6 (2)	
Not reported	7 (2)	6 (4)	1 (1)	
Clinical stage				0.507
0–II	119 (27)	41 (16)	78 (28)	
III–IV	283 (65)	99 (38)	184 (66)	
Not reported	34 (8)	118 46()	16 (6)	
Number of free flaps				0.659
One	423 (95)	155 (95)	268 (95)	
Two	22 (5)	9 (5)	13 (5)	
Flap type				0.113
Radial forearm	235 (53)	91 (55)	144 (51)	
Fibula	95 (21)	41 (25)	54 (19)	
Anterolateral thigh	57 (13)	16 (10)	41 (15)	
Other	58 (13)	16 (10)	42 (15)	
Resection extent				0.173
Soft tissue	329 (74)	113 (69)	216 (77)	
Bone	97 (22)	42 (26)	55 (20)	
Soft tissue and bone	19 (4)	9 (5)	10 (3)	

Fisher’s exact tests were used for *p*-values, except for age (* independent *t*-test).

## Data Availability

The data for this study are under the custodianship of Alberta Health Services (AHS) and are therefore unavailable for sharing. Data can be made available after an appropriate data sharing and access agreement is formally completed. Please contact Dort for more information (jdort@ucalgary.ca).

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
