# Peer review of "From Pathways to Practice: Impact of Implementing Mobilization Recommendations in Head and Neck Cancer Surgery with Free Flap Reconstruction"

_cancers, 2021, doi:10.3390/cancers13122890_

Round 1
Reviewer 1 Report
This is a retrospective single-institutional study on 445 patients that underwent major head and neck cancer surgery followed by free flap reconstruction. The investigators looked at the effects of implementing ERAS guideline recommendations, in particular early mobilization rules, and studied compliance, effects on LOS, and complication rates.
This is an important study and fits well head to head with the companion paper submitted to the same issue.
I suggest to elaborate on the uni- and multivariate analysis on the effect of "compliance with recommendation" and LOS and complication rate. Adding the results as tables would be helpful here.
Compliance with recommendation was apparently not found to be a significant predictor of LOS and complications rate in the multivariate analysis? This should be clarified and if so explained.
A reflection on the significance of a reduction of 2 days of LOS should be added to the discussion
Minor comments: Text to be corrected for repetition between line 218 and 223
Reviewer 2 Report
Dear Authors,
thank you for submission.
The manuscript is very well written, however there is a Figure 1 missing - there is only the description. Please add the missing figure.
My biggest concern is the rationale behind submitting two similar papers - in my opinion the aims, methodology and results of your two papers are so similar it should be combined in one manuscript.
Reviewer 3 Report
This is an interesting study about the impact of implementing mobilization recommendations in head and neck cancer surgery. The authors evaluated compliance with the ERAS recommendation for early postoperative mobilization in 445 consecutive patients who underwent HNC surgery with free flap reconstruction.
The paper is well written. Materials and methods are adequately described and discussion is appropriate. However, some minor issues remain.
Please specify that the patients underwent free flap reconstruction in the title.
The authors must add the types of demolitive procedures before reconstruction in Table 1.
Figure 1 is not included in the paper.
In the Discussion, the authors stated that the introduction of an opioid-sparing pain management protocol increased the proportion of patients with adequate pain control (pain rating below 4/10) by ~10% in the first 24 hours after surgery. More data about pain control should be added in the Results section.
Round 2
Reviewer 2 Report
Dear Authors,
Thank you for your explanation, I have no further comments.
Reviewer 3 Report
Thank you for improving the manuscript.